# Cohort profile: Biomarkers related to folate-dependent one-carbon metabolism in colorectal cancer recurrence and survival – the FOCUS Consortium

Biljana Gigic ●,[1] Eline van Roekel,[2] Andreana N Holowatyj,[3,4] Stefanie Brezina,[5] Anne J M R Geijsen,[6] Arve Ulvik,[7] Jennifer Ose,[8,9] Janna L Koole,[2] Victoria Damerell,[1] Rama Kiblawi,[8,9] Tanja Gumpenberger,[5] Tengda Lin,[8,9] Gry Kvalheim,[7] Torsten Koelsch,[1] Dieuwertje E Kok,[6] Franzel J van Duijnhoven,[6] Martijn J Bours,[2] Andreas Baierl,[10] Christopher I Li,[11] William Grady,[11,12] Kathy Vickers,[11] Nina Habermann,[13] Martin Schneider,[1] Ellen Kampman,[6] Per Magne Ueland,[7] Alexis Ulrich,[1,14] Matty Weijenberg,[2] Andrea Gsur,[5] Cornelia Ulrich ●,[8,9] The FOCUS Consortium

EK, PMU, AU, MW, AG and CU are joint senior authors.

For numbered affiliations see end of article.

**Correspondence to**
Professor Cornelia Ulrich; neli.ulrich@hci.utah.edu and Professor Andrea Gsur; andrea.gsur@meduniwien.ac.at

## ABSTRACT

**Purpose** The overarching goal of the FOCUS (biomarkers related to folate-dependent one-carbon metabolism in colorectal cancer (CRC) recurrence and survival) Consortium is to unravel the effect of folate and folate-mediated one-carbon metabolism (FOCM) biomarkers on CRC prognosis to provide clinically relevant advice on folate intake to cancer patients and define future tertiary prevention strategies.

**Participants** The FOCUS Consortium is an international, prospective cohort of 2401 women and men above 18 years of age who were diagnosed with a primary invasive non-metastatic (stages I–III) CRC. The consortium comprises patients from Austria, two sites from the Netherlands, Germany and two sites from the USA. Patients are recruited after CRC diagnosis and followed at 6 and 12 months after enrolment. At each time point, sociodemographic data, data on health behaviour and clinical data are collected, blood samples are drawn.

**Findings to date** An increased risk of cancer recurrences was observed among patients with higher compared with lower circulating folic acid concentrations. Furthermore, specific folate species within the FOCM pathway were associated with both inflammation and angiogenesis pathways among patients with CRC. In addition, higher vitamin $B_6$ status was associated with better quality of life at 6 months post-treatment.

**Future plans** Better insights into the research on associations between folate and FOCM biomarkers and clinical outcomes in patients with CRC will facilitate the development of guidelines regarding folate intake in order to provide clinically relevant advice to patients with cancer, health professionals involved in patient care, and ultimately further tertiary prevention strategies in the future. The FOCUS Consortium offers an excellent infrastructure for short-term and long-term research projects and for combining additional biomarkers and data resulting from the individual cohorts within the next years,

## STRENGTHS AND LIMITATIONS OF THIS STUDY

⇒ FOCUS is the largest consortium to date addressing the research question of folate and folate-mediated one-carbon metabolism biomarkers in relation to survival, recurrence, treatment toxicity and health-related quality of life outcomes in patients with colorectal cancer (CRC).

⇒ The cohorts included in the FOCUS Consortium are designed to enable future pooling of data using harmonised and standardised methods to collect data and biospecimens.

⇒ The pooled sample size provides sufficient power to investigate subgroup analyses across patients with CRC.

⇒ Study time point definitions differ between some of the cohorts and have to be adapted for specific projects.

⇒ A selection bias for follow-up can arise because it is possible that patients who experience more severe toxicities, worse clinical outcome or health-related quality of life are under-represented among those completing follow-ups.

for example, microbiome data, omics and multiomics data or CT-quantified body composition data.

## INTRODUCTION

Among men and women worldwide, colorectal cancer (CRC) accounts for nearly 10% of all incident cancer cases.[1] In Europe, 5-year survival for patients with CRC is approximately 55%, with substantial differences by stage.[2 3] The number of patients with CRC continues to increase due to implementation of improved screening strategies and/or enhanced treatment modalities.[4]

Many patients with cancer seek information on what they can do themselves to improve survival—for instance, by improving their dietary habits and other lifestyle factors.[5] However, there may be behavioural aspects among individuals diagnosed with CRC, which may be harmful in some cases, such as the use of high-dose nutritional supplements containing synthetic folate. In general, knowledge on short-term and long-term effects is insufficient to make sound recommendations on use of dietary supplements, in particular folate, to cancer survivors, even though dietary supplements are used by 20%–85% of patients with cancer,[6–8] highlighting the importance of thorough evaluation of potential benefits and harms and to support development of evidence-based recommendations on use of dietary supplements to patients with cancer.

Folic acid is the synthetic form of the B-vitamin folate and is often used in dietary supplements or fortified foods. However, there is broad agreement that food folate is less bioavailable than folic acid with a median relative bioavailability of 65% (range: 44%–80%), an estimate that approximates the 60% value derived from the dietary folate equivalents equation.[9] Folate and folic acid play an important role in one-carbon metabolism, which is a complex series of biochemical reactions essential in nucleotide synthesis, methylation reactions and amino acid homeostasis.[10] One-carbon metabolism refers to a complex network of biochemical reactions linked to nucleotide synthesis and provides methyl groups for DNA, RNA or protein methylation. Thus, one-carbon metabolism is directly controlling processes determining DNA synthesis and integrity, both processes known to be linked to tumour growth.[11] To what extent folic acid supplement use and biomarkers of folate-mediated one-carbon metabolism (FOCM) impact cancer survival and treatment efficacy and toxicity still needs to be clarified.[11 12] Folate and FOCM biomarker deficiencies may increase cancer risk, but high levels, especially of synthetic folic acid, may also be driving factors in carcinogenesis.[6 13] An increasing body of evidence suggests that folate plays a dual role in carcinogenesis, involving both the prevention of early lesions and potential harm once preneoplastic or neoplastic lesions have developed.[6 12 14–20]

The overarching goal of the FOCUS (biomarkers related to FOlate-dependent one-carbon metabolism in Colorectal cancer recUrrence and Survival) Consortium is to study associations between folate and FOCM biomarkers and recurrence and survival in patients with CRC. Better insights into these associations will facilitate the development of guidelines regarding folate intake in order to provide clinically relevant advice to cancer patients, health professionals involved in patient care, and ultimately further tertiary prevention strategies in the future. The FOCUS Consortium is a large-scale international consortium with patients with CRC from six prospective cohort studies. The primary objectives of the FOCUS Consortium are as follows: (1) to determine possible associations of folate and other FOCM biomarkers at diagnosis with recurrence or survival in non-metastatic (stages I–III) CRC; (2) to elucidate whether biomarkers related to FOCM are associated with dietary and supplemental intake of these respective nutrients; (3) to explore whether FOCM biomarkers are associated with treatment toxicity in patients with CRC undergoing chemotherapy and (4) to collect comprehensive data of patient characteristics at baseline and follow-up including biomarkers of FOCM to establish an unique resource for future scientific research. The FOCUS Consortium is funded by the European Research Area Network on Translational Cancer Research.

The main purpose of the FOCUS Cohort profile is to (1) inform the scientific community about the FOCUS Consortium, (2) describe the complex methodology of a large consortium, (3) present ongoing studies using this infrastructure and (4) advise interested researchers of opportunities for collaboration. This joint research may lead to a better understanding of the role of folate-related and FOCM-related mechanisms in the prognosis of CRC and be a precursor for data for future randomised controlled trials (RCTs), which will be critical for the development of guidelines regarding folate intake among patients with CRC.

## Cohort description

The FOCUS Consortium is an international, prospective consortium including six cohort studies recruiting women and men at the age of 18 years and older diagnosed with a primary CRC. The FOCUS Consortium is composed of patients from the ColoCare Study at the University of Heidelberg, Germany (n=298, 12.3%), the ColoCare Study at the Huntsman Cancer Institute in Salt Lake City, Utah, USA (n=80, 3.3%), the ColoCare Study at the Fred Hutchinson Cancer Research Center in Seattle, Washington, USA (n=157, 6.3%), the COLON Study (COlorectal cancer: Longitudinal, Observational study on nutritional and lifestyle factors that may influence colorectal tumour recurrence, survival and quality of life) at Wageningen University and Research in the Netherlands (n=1365, 56.1%), the CORSA Study (Colorectal Cancer Study of Austria) at the Medical University of Vienna in Austria (n=218, 9.0%) and the EnCoRe Study (Energy for life after ColoRectal cancer) at Maastricht University Medical Center+in the Netherlands (n=317, 13.0%).

In total, n=2435 patients with CRC were considered for the FOCUS Consortium, of which n=34 patients were excluded due to tumour staging being either 0 or IV leading to a total number of n=2401 stages I–III patients with CRC included in further analyses.

Patients were recruited after CRC diagnosis and repeated study measurements were conducted at time of recruitment, and at 6 and 12 months thereafter. At each study time point, sociodemographic data, data on lifestyle factors (eg, diet, supplement use, physical activity) and clinical data (eg, tumour site and stage, cancer treatment, treatment-induced toxicities, recurrence, survival)

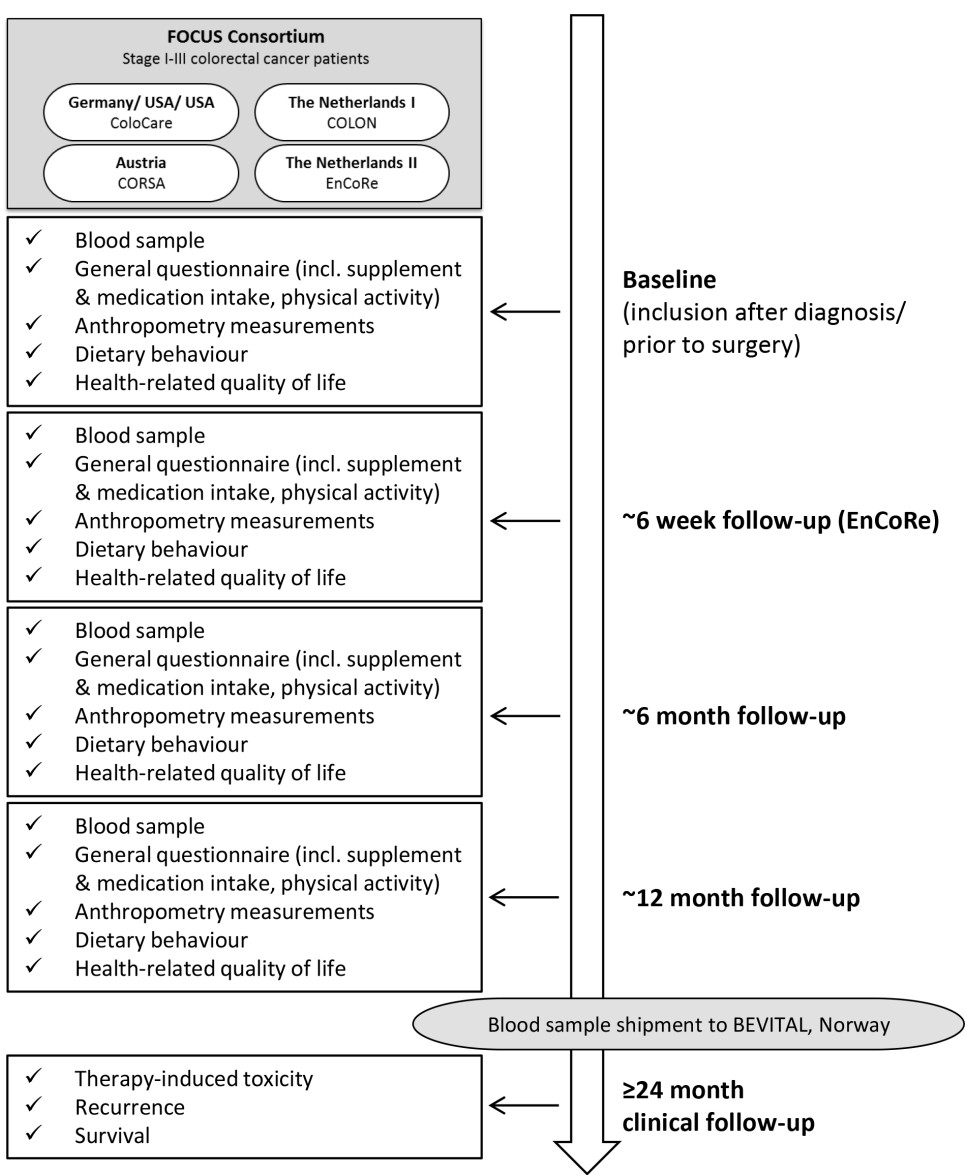

**FOCUS Consortium**
Stage I-III colorectal cancer patients

Germany/ USA/ USA
ColoCare

The Netherlands I
COLON

Austria
CORSA

The Netherlands II
EnCoRe

✓ Blood sample
✓ General questionnaire (incl. supplement & medication intake, physical activity)
✓ Anthropometry measurements
✓ Dietary behaviour
✓ Health-related quality of life

**Baseline**
(inclusion after diagnosis/ prior to surgery)

✓ Blood sample
✓ General questionnaire (incl. supplement & medication intake, physical activity)
✓ Anthropometry measurements
✓ Dietary behaviour
✓ Health-related quality of life

**~6 week follow-up (EnCoRe)**

✓ Blood sample
✓ General questionnaire (incl. supplement & medication intake, physical activity)
✓ Anthropometry measurements
✓ Dietary behaviour
✓ Health-related quality of life

**~6 month follow-up**

✓ Blood sample
✓ General questionnaire (incl. supplement & medication intake, physical activity)
✓ Anthropometry measurements
✓ Dietary behaviour
✓ Health-related quality of life

**~12 month follow-up**

Blood sample shipment to BEVITAL, Norway

✓ Therapy-induced toxicity
✓ Recurrence
✓ Survival

**≥24 month clinical follow-up**

**Figure 1** FOCUS Consortium design.

were collected, and blood samples were drawn (figure 1). Below, a more specific description of each included study is provided.

The ColoCare Study

The ColoCare Study (ClinicalTrials.gov identifier: NCT02328677) is an ongoing international, multicentre prospective cohort study among women and men newly diagnosed with a primary invasive CRC, with the goal to investigate predictors of cancer recurrence, survival, treatment toxicities and health-related quality of life.[4 21] Three ColoCare Consortium sites participate in the FOCUS Consortium: the Fred Hutchinson Cancer Research Center in Seattle (Washington, USA), and the University Hospital Heidelberg, Heidelberg (Germany) as well as the Huntsman Cancer Institute in Salt Lake City (Utah, USA). Patients are enrolled prior to undergoing CRC surgery according to the following inclusion criteria: individuals who are 18 years of age and older, newly diagnosed (ie,

non-recurrent) with invasive CRC. Blood draws and other biospecimens are obtained prior to surgery and at regular intervals (eg, 6, 12, 24 months). Questionnaires are administered to assess lifestyle behaviour, health-related quality of life and clinical outcomes such as CRC recurrence, treatment and treatment symptoms at each study time point. Clinical data are obtained through reviews of patient medical records, pathology and imaging reports. Vital status is obtained through medical records, routine follow-up mailings, periodic requests for outside medical records and state or national cancer and death registries.

The COLON Study

The COLON Study (ClinicalTrials.gov identifier: NCT03191110) started in 2010 and is an ongoing, multicentre prospective cohort study specifically designed to assess associations between nutrition, lifestyle and dietary supplement use with quality of life, CRC recurrence and survival among patients with CRC (stages I–IV).[22] Persons

with a history of CRC or (partial) bowel resection, chronic inflammatory bowel disease, or a known hereditary CRC syndrome are excluded from the study. Patients are recruited from 11 regional and academic hospitals prior to surgery. Individuals donate blood samples and provide information on diet, lifestyle and dietary supplement use at CRC diagnosis, that is, baseline, and at 6, 12 (chemotherapy patients only), 24 and 60 months after diagnosis. Clinical data are collected through review of medical records (treatment-induced toxicity) or through linkage with the Dutch ColoRectal Audit (DCRA). Mortality data (ie, death or alive and date of death) are retrieved from the Municipal Personal Records Database. Recurrence data have been retrieved in collaboration with the Netherlands Cancer Registry.

### The CORSA Study

CORSA is an ongoing case–control study of women and men recruiting patients with high-risk and low-risk adenomas and population-based colonoscopy negative controls, with an age range between 30 and 90 years. Since 2003, more than 13 500 participants have been recruited across nine sites in Austria. The multicentre recruitment within CORSA follows standardised protocols, resulting in consistent data from all recruitment sites. These sites include the Medical University of Vienna (Department of Surgery), two hospitals in Vienna (Clinic Favoriten and Clinic Landstraße) and the Hospital Wiener Neustadt in Lower Austria. Furthermore, the recruitment for CORSA was performed in four hospitals in the federal state Burgenland within the population-based screening programme 'Burgenland PREvention Trial of colorectal cancer DIsease with ImmunologiCal Testing' (B-PRE-DICT). B-PREDICT is a two-stage screening project where more than 150 000 inhabitants of Burgenland aged between 40 and 80 are invited annually to participate in this programme using a faecal immunochemical test (FIT) as an initial screening. FIT-positive (≥10 µg haemoglobin/g faeces) tested individuals are offered a complete colonoscopy and are asked to participate in CORSA. Biospecimen are collected at each site using harmonised protocols. CORSA participants provide a basic CORSA questionnaire assessing data on body mass index (BMI), smoking history, alcohol consumption, education level, family status, profession, basic dietary habits and diabetes. Clinical data are abstracted from medical records and processed in a structured database following standardised documentation guidelines and according to the General Data Protection Regulation.

### The EnCoRe Study

The EnCoRe Study is a prospective cohort study initiated in 2012 at the Maastricht University Medical Center+in the Netherlands that focuses on the importance of lifestyle factors for quality of life, recurrence and survival of patients with CRC.[23] The EnCoRe Study is registered in the Netherlands Trial Registry for experimental and observational studies (www.trialregister.

nl, registration number 7099). Stages I–III patients with CRC are enrolled at diagnosis at three hospitals in the South-Eastern region of the Netherlands and followed up until 5 years after completion of treatment with repeated measurements at diagnosis (pretreatment), and at 6 weeks and 6, 12, 24 and 60 months after the end of the initial anticancer treatment (ie, surgery, chemotherapy, radiotherapy). Patients with stage IV CRC, an inability to understand the Dutch language in speech or writing, with comorbidities obstructing successful participation (eg, Alzheimer disease), or with severe visibility or hearing disorders are excluded from the study. Repeated home visits by trained dieticians are conducted and data are collected among others on sociodemographic factors, quality of life, functioning, physical activity, comorbidity, dietary intake, supplement use and anthropometry. In addition, clinical data are collected from hospital records and blood samples are drawn at all time points. Recurrence data have been retrieved in collaboration with the Netherlands Cancer Registry. Mortality data (ie, death or alive and date of death) are retrieved from the Municipal Personal Records Database.

### The FOCUS Consortium

Sociodemographic and clinical characteristics of the 2401 participants in the FOCUS Consortium are presented in table 1. The mean age at CRC diagnosis was 65.4 years (SD: 10.2; range: 22–93 years), and the majority of participants were male (64.0%). The mean BMI was 27.1 kg/m² (SD: 4.6).

In terms of education status, 35.9% of the patients reported lower education, 28.9% reported intermediate education and 27.4% high education. Most of the patients were married or part of a living community (73.2%). About half of the overall cohort reported to be a former smoker (50.8%), 31.9% were never-smokers and 12.1% were current smokers (5.2% were unknown). Mean alcohol intake was 13.6 g/day (SD: 17.1). In total, 16.9% of the patients reported regular dietary supplementation (ie, at least once per week during the last 4 weeks) of folic acid. Over half of all participants (54.5%) reported adherence to the physical activity guidelines of at least 150 min per week of moderate-to-vigorous physical activity. Lifestyle characteristics of the FOCUS cohort are presented in table 2.

Regarding clinical characteristics, 60.4% of participants were diagnosed with colon cancer, 35.9% with rectal cancer, and 1.7% with rectosigmoid cancer (2.0% were of unknown tumour subsite). In total, 26.2% were diagnosed with stage I, 28.9% with stage II and 40.5% with stage III CRC. Approximately 1% of participants were classified with an unspecified cancer stage, as distinction between stage I and II or II and III was not possible, and for 3.7% of the total population the cancer stage was unknown. In total, 95.4% of patients underwent surgery, whereas neoadjuvant chemotherapy was administered to 13.8% of patients and 28.4% received adjuvant chemotherapy. Neoadjuvant radiotherapy was administered to

**Table 1**  Sociodemographic and clinical factors of eligible focus Consortium participants at baseline (n=2401)

| | Overall | ColoCare HD | ColoCare FHCRC | ColoCare HCI | COLON | CORSA | EnCoRe |
|---|---|---|---|---|---|---|---|
| n (%) | 2401 (100) | 298 (12.4) | 150 (6.3) | 80 (3.3) | 1338 (55.7) | 218 (9.1) | 317 (13.2) |
| **Age** | | | | | | | |
| Mean year±SD | 65.4±10.2 | 63.7±12.2 | 58.3±12.8 | 61.3±11.2 | 66.1±8.7 | 67.3±12.0 | 66.9±9.3 |
| **Sex** | | | | | | | |
| Female n (%) | 864 (36.0) | 100 (33.6) | 67 (44.7) | 32 (40.0) | 486 (36.3) | 76 (34.9) | 103 (32.5) |
| Male n (%) | 1537 (64.0) | 198 (66.4) | 83 (55.3) | 48 (60.0) | 852 (63.7) | 142 (65.1) | 214 (67.5) |
| **Education** | | | | | | | |
| Low n (%) | 861 (35.9) | 123 (41.3) | 18 (12.0) | 17 (21.3) | 543 (40.6) | 73 (33.5) | 87 (27.4) |
| Intermediate n (%) | 694 (28.9) | 57 (19.1) | 71 (47.3) | 40 (50.0) | 332 (24.8) | 73 (33.5) | 121 (38.2) |
| High n (%) | 658 (27.4) | 88 (29.5) | 28 (18.7) | 8 (10.0) | 417 (31.7) | 14 (6.4) | 103 (32.5) |
| Unknown n (%) | 188 (7.8) | 30 (10.1) | 33 (22.0) | 15 (18.7) | 46 (3.4) | 58 (26.6) | 6 (1.9) |
| **Marital status** | | | | | | | |
| Unmarried n (%) | 122 (5.1) | 19 (6.4) | 15 (10.0) | 2 (2.5) | 60 (4.5) | 10 (4.6) | 16 (5.1) |
| Married n (%) | 1738 (72.4) | 193 (64.8) | 71 (47.3) | 50 (62.5) | 1071 (80.0) | 113 (51.8) | 240 (75.7) |
| Divorced/separated n (%) | 138 (5.7) | 24 (8.0) | 11 (7.3) | 10 (12.5) | 64 (4.8) | 14 (6.4) | 15 (4.7) |
| Widowed n (%) | 194 (8.1) | 31 (10.4) | 7 (4.7) | 1 (1.3) | 90 (6.7) | 26 (11.9) | 39 (12.3) |
| Living community n (%) | 20 (0.8) | 0 | 13 (8.7) | 2 (2.5) | 4 (0.3) | 1 (0.5) | 0 |
| Unknown n (%) | 189 (8.9) | 31 (10.4) | 33 (22.0) | 15 (18.7) | 49 (3.7) | 54 (24.8) | 7 (2.2) |
| **Smoking status** | | | | | | | |
| Current n (%) | 290 (12.1) | 53 (17.8) | 8 (5.3) | 6 (7.5) | 143 (10.7) | 38 (17.4) | 42 (13.3) |
| Former n (%) | 1219 (50.8) | 132 (44.3) | 53 (35.32) | 21 (26.3) | 767 (57.3) | 76 (34.9) | 170 (53.6) |
| Never n (%) | 767 (31.9) | 90 (30.2) | 56 (37.3) | 39 (48.7) | 386 (28.9) | 97 (44.5) | 99 (31.2) |
| Unknown n (%) | 125 (5.2) | 23 (7.7) | 33 (22.0) | 14 (17.5) | 42 (3.1) | 7 (3.2) | 6 (1.9) |
| **Alcohol intake** | | | | | | | |
| Mean g/day±SD | 13.6±17.1 | 16.1±19.2 | 8.8±18.5 | 2.3±10.2 | 14.0±17.0 | | 12.9±16.5 |
| Unknown n (%) | 633 (26.4) | 208 (69.8) | 81 (54.0) | 60 (75.0) | 54 (4.1) | 218 (100) | 12 (3.8) |
| **BMI at diagnosis** | | | | | | | |
| Mean kg/m$^2$±SD | 27.1±4.6 | 26.6±4.1 | 28.7±7.3 | 29.4±7.5 | 26.6±4.0 | 27.7±4.4 | 28.3±4.6 |
| Unknown n (%) | 42 (1.7) | 2 (0.7) | 0 | 14 (17.5) | 10 (0.8) | 14 (6.4) | 2 (0.6) |
| **Tumour site** | | | | | | | |
| Colon n (%) | 1450 (60.4) | 140 (57.0) | 75 (50.0) | 45 (56.3) | 867 (64.8) | 131 (60.1) | 192 (60.6) |
| Rectum n (%) | 862 (35.9) | 143 (48.0) | 60 (40.0) | 22 (27.5) | 434 (32.4) | 78 (35.8) | 125 (39.4) |
| Rectosigmoid n (%) | 42 (1.7) | 15 (5.0) | 15 (10.0) | 7 (8.7) | 0 | 5 (2.3) | 0 |
| Unknown n (%) | 47 (2.0) | 0 | 0 | 6 (7.5) | 37 (2.8) | 4 (1.8) | 0 |
| **Tumour stage** | | | | | | | |
| I n (%) | 629 (26.2) | 76 (25.5) | 35 (23.3) | 21 (26.3) | 326 (24.4) | 82 (37.6) | 89 (28.1) |
| II n (%) | 694 (28.9) | 114 (38.3) | 49 (32.7) | 23 (28.7) | 386 (28.9) | 52 (23.9) | 70 (22.1) |
| III n (%) | 973 (40.5) | 108 (36.2) | 66 (44.0) | 36 (45.0) | 564 (42.1) | 52 (23.9) | 147 (46.4) |
| Unspecified* n (%) | 16 (0.7) | 0 | 0 | 0 | 4 (0.3) | 12 (5.5) | 0 |
| Unknown n (%) | 89 (3.7) | 0 | 0 | 0 | 58 (4.3) | 20 (9.2) | 11 (3.5) |
| **Treatment** | | | | | | | |
| Neoadjuvant chemotherapy n (%) | 331 (13.8) | 55 (18.5) | 47 (31.3) | 15 (18.8) | 138 (10.3) | 14 (6.4) | 62 (19.6) |
| Unknown n (%) | 58 (2.4) | 2 (0.7) | 1 (0.7) | 10 (12.5) | 38 (2.8) | 7 (3.2) | 0 |
| Neoadjuvant radiotherapy n (%) | 523 (21.8) | 72 (24.2) | 48 (32.0) | 13 (16.3) | 290 (21.7) | 14 (6.4) | 86 (27.1) |
| Unknown n (%) | 58 (2.4) | 2 (0.7) | 1 (0.7) | 10 (12.5) | 38 (2.8) | 7 (3.2) | 0 |
| Adjuvant chemotherapy n (%) | 682 (28.4) | 100 (33.6) | 80 (53.3) | 32 (40.0) | 315 (23.8) | 56 (25.7) | 99 (31.2) |
| Unknown n (%) | 121 (5.0) | 17 (5.7) | 3 (2.0) | 28 (35.0) | 63 (4.7) | 9 (4.1) | 1 (0.3) |

**Table 1** Continued

|  | Overall | ColoCare HD | ColoCare FHCRC | ColoCare HCI | COLON | CORSA | EnCoRe |
|---|---|---|---|---|---|---|---|
| Adjuvant radiotherapy n (%) | 227 (9.5) | 188 (63.1) | 7 (4.7) | 1 (1.3) | 16 (1.2) | 13 (6.0) | 2 (0.6) |
| Unknown n (%) | 164 (6.8) | 17 (5.7) | 3 (2.0) | 28 (35.0) | 106 (7.9) | 9 (4.1) | 1 (0.3) |
| Surgery n (%) | 2291 (95.4) | 298 (100.0) | 149 (99.3) | 71 (88.7) | 1290 (96.4) | 195 (89.5) | 288 (90.9) |
| Unknown n (%) | 44 (1.8) | 0 | 0 | 3 (3.8) | 39 (2.9) | 1 (0.5) | 1 (0.3) |

*Distinguishing between stage I and II or II and III not possible.
BMI, body mass index.

21.8% of patients and adjuvant radiotherapy in 9.5% of the population.

### Patient and public involvement

Patients and the public were not involved in the design, conduct, reporting or dissemination plans of this research.

### Cohort follow-up

At each study site, collection of biospecimen, clinical, demographic, questionnaire and anthropometric data occurred at baseline, and at 6 and 12 months following recruitment. Baseline measurements in CORSA, COLON and EnCoRe have been performed at diagnosis (preferably prior to any cancer treatment) while in the ColoCare Study such measurements have been done prior to surgery (ie, neoadjuvant treatment might have been applied prior to baseline blood and data collection).

Blood samples at baseline were collected from n=2132 (88.8%) study participants. N=1537 (64.0%) participants donated blood at the 6 months and n=614 (25.6%) at the 12 months follow-up time point. In addition, n=251 (10.5%) EnCoRe patients provided blood samples at 6 weeks post-treatment, a subset specifically of interest at the Maastricht study site.

### Data collection

Data collection at baseline and follow-up time points is summarised in table 3 and is briefly described below.

### Lifestyle and demographic data

ColoCare questionnaires and standardised Food Frequency Questionnaires (FFQ) are used to assess intake of dietary supplements and medication, smoking, dietary intake and other health behaviours at each study time point. In COLON, patients provide information on diet, lifestyle and supplement use via COLON questionnaires and FFQs. Patients from the CORSA Study are requested to complete a questionnaire assessing anthropometric and demographic factors. Patients enrolled in EnCoRe receive repeated home visits by trained dieticians, where extensive measurements are performed that include assessment of demographic data, physical activity (both questionnaire and accelerometry based) and smoking behaviour (questionnaire data), dietary intake (FFQs at diagnosis and 7-day food diaries at follow-up measurements), supplement use (registered by dieticians) and anthropometry measurements.

### Clinical data and outcomes: medical chart abstraction

All cancers and medical diagnoses are classified according to International Classification of Diseases, 10th Revision codes. Details on CRC treatment, including type of treatment regimens and treatment toxicity, are abstracted via medical records for all cohorts. Detailed information on the primary outcomes of interest, CRC recurrence and survival, are ascertained through reviews of medical records, pathology reports and imaging reports documenting the diagnosis of a recurrence. Data on recurrence and vital status is supplemented from the clinical cancer registries and survival data are verified by the inhabitant registries that exist at all study sites. For COLON and EnCoRe, all data on recurrence are retrieved in collaboration with the Netherlands Cancer Registry. Moreover, survival data for CORSA participants were obtained by the Main Association of Austrian Social Insurance Carriers as well as from Statistics Austria.

### Patient-reported outcomes

Health-related quality of life is assessed by the validated and widely used cancer-specific 30-item core questionnaire QLQ-C30 and the 29-item CRC module QLQ-CR29, developed by the European Organisation for Research and Treatment of Cancer (EORTC).[24] Chemotherapy-induced peripheral neuropathy (CIPN), which is a common toxicity in patients with CRC, is measured by the EORTC QLQ-CIPN20.[25] Patient-reported outcomes are available for ColoCare Heidelberg, COLON and EnCoRe.

### Biomarkers of FOCM

Blood is processed in identical settings across all study centres. Plasma (CORSA, COLON and EnCoRe) and serum (for the three ColoCare Study sites) samples were collected and immediately centrifuged, aliquoted and stored at −80°C.

All biological analyses were performed at BEVITAL AS (Bergen, Norway, http://www.bevital.no), which carried out metabolic profiling of biomarkers allocated to seven complementary analytical platforms. Apart from analyses of microbiological active folate[26] and vitamin $B_{12}$[27]), all analyses were based on mass spectrometry. Circulating folate, separate folate species, 5-methyltetrahydrofolate (the main form in plasma), folic acid (folate in supplements), 4a-OH 5-methyltetrahydrofolate (the primary oxidation form of circulating folate), folate catabolites

**Table 2** Lifestyle characteristics of eligible focus Consortium participants at baseline (n=2401)

| | Overall | ColoCare HD | ColoCare FHCRC | ColoCare HCl | COLON | CORSA | EnCoRe |
|---|---|---|---|---|---|---|---|
| n (%) | 2401 (100) | 298 (12.4) | 150 (6.3) | 80 (3.3) | 1338 (55.7) | 218 (9.1) | 317 (13.2) |
| Folic acid supplementation | | | | | | | |
| Yes n (%) | 405 (16.9) | 4 (1.3) | 9 (5.9) | 2 (2.5) | 325 (24.3) | | 65 (20.5) |
| Unknown n (%) | 383 (15.9) | 27 (9.1) | 86 (56.6) | 14 (17.5) | 38 (2.8) | 218 (100) | 0 |
| Vitamin $B_{12}$ supplementation | | | | | | | |
| Yes n (%) | 430 (17.9) | 10 (3.4) | 8 (5.3) | 14 (17.5) | 336 (25.1) | | 62 (19.6) |
| Unknown n (%) | 384 (16.0) | 27 (9.1) | 86 (56.6) | 14 (17.5) | 39 (2.9) | 218 (100) | 0 |
| Vitamin $B_6$ supplementation | | | | | | | |
| Yes n (%) | 77 (3.2) | 8 (2.7) | 4 (2.6) | 4 (5.0) | | | 61 (19.2) |
| Unknown n (%) | 1684 (70.1) | 27 (9.1) | 86 (56.6) | 14 (17.5) | 1339 (100) | 218 (100) | 0 |
| Vitamin $B_2$ supplementation | | | | | | | |
| Yes n (%) | 62 (2.6) | 0 | 2 (1.3) | 0 | | | 60 (18.9) |
| Unknown n (%) | 1684 (70.1) | 27 (9.1) | 86 (56.6) | 14 (17.5) | 1339 (100) | 218 (100) | 0 |
| Dietary intake of folate equivalents | | | | | | | |
| Mean µg/day±SD | 227.1±94.3 | 115.1±38.1 | 149.6±97.5 | 116.2±112.8 | 229.0±88.7 | | 276.3±82.2 |
| Unknown n (%) | 638 (26.5) | 208 (69.8) | 83 (54.6) | 60 (75.0) | 57 (4.3) | 218 (100) | 12 (3.8) |
| Dietary intake of vitamin $B_{12}$ | | | | | | | |
| Mean µg/day±SD | 4.6±2.4 | 6.2±2.9 | 7.1±5.8 | 4.2±2.2 | 4.3±2.1 | | 4.6±2.0 |
| Unknown n (%) | 638 (26.5) | 208 (69.8) | 83 (54.6) | 60 (75.0) | 57 (4.3) | 218 (100) | 12 (3.8) |
| Dietary intake of vitamin $B_6$ | | | | | | | |
| Mean mg/day±SD | 1.5±0.5 | 1.7±0.5 | 2.0±0.7 | 1.5±0.7 | 1.5±0.5 | | 1.8±0.5 |
| Unknown n (%) | 638 (26.5) | 208 (69.8) | 83 (54.6) | 60 (75.0) | 57 (4.3) | 218 (100) | 12 (3.8) |
| Dietary intake of vitamin $B_2$ | | | | | | | |
| Mean mg/day±SD | 1.4±0.5 | 1.5±0.5 | 2.2±1.0 | 1.6±0.8 | 1.3±0.4 | | 1.4±0.5 |
| Unknown n (%) | 638 (26.5) | 208 (69.8) | 83 (54.6) | 60 (75.0) | 57 (4.3) | 218 (100) | 12 (3.8) |
| Total energy intake | | | | | | | |
| Mean kcal/day±SD | 1941.8±593.6 | 2330.9±714.0 | 1835.0±775.7 | 1398.5±597.8 | 1856.9±517.2 | | 2243.7±651.0 |
| Unknown n (%) | 638 (26.5) | 208 (69.8) | 83 (54.6) | 60 (75.0) | 57 (4.3) | 218 (100) | 12 (3.8) |
| Adherence to physical activity guidelines* | | | | | | | |
| Yes n (%) | 1310 (54.5) | 84 (28.2) | 35 (23.0) | 20 (25.0) | 941 (70.3) | | 230 (72.6) |
| Unknown n (%) | 370 (15.4) | 33 (11.1) | 53 (34.9) | 15 (18.7) | 45 (3.4) | 218 (100) | 6 (1.9) |

*Self-reported engagement in at least 150 min per week of moderate-to-vigorous physical activity.

**Table 3**  Variables available at baseline, and at 6 and 12 months follow-up within the FOCUS Consortium

| Category | Variables | Baseline | 6 months | 12 months |
|---|---|---|---|---|
| Demographics | Age | x | x | x |
| | Gender | x | | |
| | Highest education | x | | |
| | Social status | x | | |
| | Height | x | | |
| | Weight | x | x | x |
| | BMI | x | x | x |
| | Smoking status | x | x | x |
| | Smoking duration | x | x | x |
| | Smoking pack years | x | x | x |
| | Menopausal status | x | | |
| | Postmenopausal hormone use | x | | |
| | Race | x | | |
| Cancer characteristics | Cancer site | x | | |
| | Cancer stage | x | | |
| | TNM classification | x | | |
| Treatment | Preoperative chemotherapy | x | | |
| | Preoperative radiotherapy | x | | |
| | Surgery | x | x | |
| | Postoperative chemotherapy | | x | x |
| | Postoperative radiotherapy | | x | x |
| Supplement intake | Folic acid supplements | x | x | x |
| | Vitamin $B_2$ supplements | x | x | x |
| | Vitamin $B_6$ supplements | x | x | x |
| | Vitamin $B_{12}$ supplements | x | x | x |
| | Vitamin A supplements | x | x | x |
| | Vitamin C supplements | x | x | x |
| | Vitamin D supplements | x | x | x |
| (Supplement intake | Vitamin E supplements | x | x | x |
| | Calcium supplements | x | x | x |
| | Magnesium supplements | x | x | x |
| | Iron supplements | x | x | x |
| | Multivitamins | x | x | x |
| Dietary nutrients | Folate equivalents | x | x | x |
| | Vitamin $B_2$ | x | x | x |
| | Vitamin $B_6$ | x | x | x |
| | Vitamin $B_{12}$ | x | x | x |
| | Vitamin A | x | x | x |
| | Vitamin C | x | x | x |
| | Vitamin D | x | x | x |
| | Vitamin E | x | x | x |
| | Total protein | x | x | x |
| | Total fat | x | x | x |
| | Total carbohydrate | x | x | x |
| Dietary nutrients | Fibre | x | x | x |
| | Saturated fatty acids | x | x | x |
| | Monounsaturated fatty acids | x | x | x |

**Table 3** Continued

| Category | Variables | Baseline | 6 months | 12 months |
|---|---|---|---|---|
| | Polyunsaturated fatty acids | x | x | x |
| | Alcohol | x | x | x |
| | Total energy | x | x | x |
| Physical activity | Light physical activity | x | x | x |
| | Moderate physical activity | x | x | x |
| | Vigorous physical activity | x | x | x |
| | Adherence to physical activity guidelines | x | x | x |
| Medical history | Diabetes mellitus | x | x | x |
| | Asthma, chronical bronchitis, COPD, emphysema | x | | |
| | Heart attack, heart failure | x | | |
| | Hypertension | x | | |
| | Stroke | x | | |
| | Ulcer of stomach or duodenum | x | | |
| | Hypothyroidism/hyperthyroidism | x | | |
| | Systemic lupus erythematosus | x | | |
| | Inflammatory bowel disease (Crohn's disease, ulcerative colitis) | x | | |
| | Familial adenomatous polyposis | x | | |
| Medical history | Lynch syndrome (hereditary nonpolyposis colorectal cancer) | x | | |
| Regular medication use | Aspirin | x | x | x |
| | NSAID | x | x | x |
| | Ibuprofen | x | x | x |
| | Naproxen | x | x | x |
| | Celecoxib/etoricoxib | x | x | x |
| Health-related quality of life | EORTC QLQ-C30 | x | x | x |
| | EORTC QLQ-CR29 | x | x | x |
| | EORTC QLQ-CIPN20 | x | x | x |
| General information | Date of questionnaire completion | x | x | x |
| | Date of blood collection | x | x | x |
| | Freeze-thaw cycles of blood samples | x | x | x |
| | Haemolysis | x | x | x |
| | Time between blood draw and processing/storage | x | x | x |

BMI, body mass index; C30, core 30-items; CIPN20, chemotherapy-induced polyneuropathy 20-items.; COPD, Chronic Obstructive Pulmonary Disease; CR29, colon/rectum 29-items; EORTC, European Organisation for Research and Treatment of Cancer; NSAID, non-steroidal anti-inflammatory drug; QLQ, quality of life questionnaire; TNM, tumor, nodes, and metastases.

(p-aminobenzoylglutamate and acetyl p-aminobenzoylglutamate),[28] $B_6$, $B_1$ and $B_3$ vitamins, kynurenines, cotinine, trans-3'-hydroxycotinine, trigonelline,[29] choline and its metabolites, creatinine, methylhistidines, arginine and methylated arginines[30] were analysed by liquid chromatography-tandem mass spectroscopy, whereas other amino acids, in addition to total homocysteine, total cysteine and methylmalonic acid were analysed by gas chromatography-tandem mass spectroscopy.[31] hsCRP and cystatin C and its variants were measured by MALDI-TOF mass spectroscopy.[32]

A comprehensive overview of the panel of biomarkers measured in blood samples of patients enrolled in the FOCUS Consortium is provided in table 4.

### Findings to date
The FOCUS Consortium provides a unique opportunity to conduct comprehensive research on folate and FOCM

**Table 4** Measured metabolites and biomarkers within the FOCUS Consortium

| Folate and one-carbon metabolites | Abbreviation | Description |
|---|---|---|
| Anthranilic acid | AA | Tryptophan metabolite |
| Asymmetric dimethylarginine | ADMA | Inhibitor of nitric oxide synthase |
| α-Ketoglutaric acid | aKG | Keto acid of the Krebs cycle |
| Alanine | Ala | Amino acid |
| Acetamidobenzoylglutamate | apABG | Folate catabolite |
| Arginine | Arg | Amino acid |
| Asparagine | Asn | Amino acid |
| Aspartic acid | Asp | Amino acid |
| Betaine | Betaine | Methyl donor |
| Choline | Choline | Methyl donor |
| C reactive protein | hsCRP | Inflammatory marker |
| Cystatin C-desS | CnCds | Cystatin C isoform |
| Cystatin C-desSSP | CnDcssp | Cystatin C isoform |
| Cystatin C | CnCn | Marker of kidney function |
| 3Pro-OH cystatin C | CnCo | Cystatin C isoform |
| 3Pro-OH cystatin C-desS | CnCods | Cystatin C isoform |
| Total concentration of detected cystatin C isoforms | CnCt | Marker of kidney function |
| Cobalamin | $B_{12}$ | Vitamin $B_{12}$ |
| Cotinine | Cot | Nicotine metabolite |
| Creatinine | Creat | Marker of kidney function |
| C reactive protein | CRP | Inflammation |
| Cystathionine | Cysta | Thioether, transsulfuration intermediate |
| Dimethylglycine | DMG | Betaine metabolite |
| Folic acid | $B_9$ | Synthetic form of folate |
| Flavin mononucleotide | FMN | $B_2$ vitamin |
| 5-Formyl-tetrahydrofolate | fTHF | Folate species |
| Glutamine | Gln | Amino acid |
| Glutamic acid | Glu | Amino acid |
| Glycine | Gly | Amino acid |
| 3-Hydroxyanthranilic acid | HAA | Tryptophan metabolite |
| Homoarginine | hArg | Amino acid |
| Histidine | Hist | Tryptophan metabolite |
| 3-Hydroxykynurenine | HK | Tryptophan metabolite |
| 4-Alfa-hydroxy-5-methyl-THF | hmTHF | Folate oxidation product |
| Isoleucine | Ile | Amino acid |
| Kynurenic acid | KA | Tryptophan metabolite |
| Kynurenine | Kyn | Tryptophan metabolite |
| Leucine | Leu | Amino acid |
| Lysine | Lys | Amino acid |
| 3-Methylhistidine (3-MH) | m3His | Marker of muscle degradation and meat intake |
| 1-Methylhistidine (1-MH) | m1His | Marker of muscle degradation and meat intake |
| Methionine | Met | Amino acid |
| Methionine sulfoxide | MetSo | Oxidation product of Met |
| Methylmalonic acid | MMA | Marker of $B_{12}$ status |
| N1-methylnicotinamide | mNAM | $B_3$ vitamin |

**Table 4** Continued

| Folate and one-carbon metabolites | Abbreviation | Description |
|---|---|---|
| 5-Methyl-tetrahydrofolate | mTHF | Folate species |
| Nicotinic acid | NA | $B_3$ vitamin |
| Nicotinamide | NAM | $B_3$ vitamin |
| Neopterin | Neopt | Inflammatory marker |
| Trans-3'-hydroxycotinine | OHCot | Nicotine metabolite |
| Ornithine | Orn | Amino acid |
| 4-Pyridoxic acid | PA | Vitamin $B_6$ catabolite |
| Para-aminobenzoylglutamate | pABG | Folate catabolite |
| Phenylalanine | Phe | Amino acid |
| Picolinic acid | Pic | Tryptophan metabolite |
| Pyridoxal | PL | $B_6$ vitamin |
| Pyridoxal 5'-phosphate | PLP | $B_6$ vitamin |
| Pyridoxine | PN | Synthetic form of vitamin $B_6$ |
| Proline | Pro | Amino acid |
| Quinolinic acid | QA | Tryptophan metabolite |
| Riboflavin | Ribo | Main circulating $B_2$ form |
| Symmetric dimethylarginine | SDMA | Marker of renal function |
| Serine | Ser | Amino acid |
| Folate | spFolate | Microbiologically active folate |
| Total cysteine | tCys | Amino acid |
| Homocysteine | tHcy | Marker of folate and $B_{12}$ status |
| Thiamine | Thi | $B_1$ vitamin |
| Threonine | Thr | Amino acid |
| Trimethylamineoxide | TMAO | Choline metabolite |
| Trimethyllysine | TML | Amino acid |
| Thiamine monophosphate | TMP | $B_1$ vitamin |
| Trigonelline | Trig | Marker of coffee consumption |
| Tryptophan | Trp | Amino acid |
| Valine | Val | Amino acid |
| Leucine | Leu | Amino acid |
| Xanthurenic acid | XA | Tryptophan metabolite |
| Kyn/Trp ratio | KTR | Marker of immune activation |
| HK/XA ratio | | Marker of $B_6$ status |

biomarkers status in the tertiary prevention of CRC using data collected both at diagnosis and during standardised follow-up time points. This well-characterised study design provides sufficient statistical power to discern prospective associations with relevant clinical outcomes, including CRC recurrence and survival, within relevant subgroups.

### Key findings and publications

We investigated associations of circulating concentrations of folate, folic acid and folate catabolites pABG and apABG, measured around time of diagnosis, with recurrence and survival among 2024 patients diagnosed with stages I–III CRC within the international FOCUS Consortium. We did not observe any statistically significant associations for folate, pABG and apABG concentrations. However, an increased risk of cancer recurrences was observed among patients with higher compared with lower circulating folic acid concentrations.[33] Further, Kiblawi et al measured associations between one-carbon metabolites, inflammation-related and angiogenesis-related biomarkers in a cross-sectional analysis of 238 patients from the ColoCare Heidelberg cohort. The study showed that specific folate species within the one-carbon metabolism pathway are associated with both inflammation and angiogenesis pathways among patients with CRC. In particular, vitamin $B_6$ species, pyridoxal 5'-phosphate (PLP), pyridoxal (PL) and pyridoxic acid (PA), were

inversely associated with inflammatory biomarkers CRP, serum amyloid A, IL-6 and IL-8. Thiamine and thiamine monophosphate were inversely correlated with the CRP and IL-6. In addition, positive correlations of PA, PL and PLP with angiogenesis biomarker VEGF-D were observed. Our findings reinforce the notion that B vitamins involved in the one-carbon metabolism may be correlated with carcinogenic processes.[34] This and further research will support the evidence base needed for the development of dietary guidelines for patients with CRC.

Further, we investigated circulating concentrations of nine biomarkers related to the B-vitamins folate, riboflavin, vitamin $B_6$ and cobalamin, measured at diagnosis and 6 months postdiagnosis, in association with health-related quality of life as assessed by the EORTC QLQ-C30 questionnaire 6 months post-treatment in three FOCUS cohorts (ColoCare Heidelberg, EnCoRe and COLON).[35] Higher PLP concentrations were cross-sectionally associated with better physical, role and social functioning, and reduced fatigue 6 months postdiagnosis. Higher HKr (3-hydroxykynurenine:(kynurenic acid+xanthurenic acid+3-hydroxyanthranilic acid+anthranilic acid)), an inverse marker of vitamin $B_6$ status, was cross-sectionally associated with worse global quality of life, and lower physical and role functioning. Dose–response relations were observed for PLP with global quality of life, physical, role and social functioning. No associations were observed for changes in biomarker concentrations between diagnosis and 6 months with quality of life outcomes. We, therefore, concluded that higher vitamin $B_6$ status was associated with better quality of life at 6 months post-treatment and that further study is needed to clarify the role of vitamin $B_6$ in relation to quality of life.

### Previous relevant findings from individual cohorts within the consortium
To date, individual cohorts from the FOCUS Consortium have initiated the examination of dietary supplement use and dietary habits over time. Among patients with CRC enrolled in the ColoCare Study the proportion of supplement users was found to be highest post-diagnosis (35%).[36] Moreover, within an international investigation including ColoCare participants from multiple sites, Ulrich *et al* showed differences in plasma folate concentration between Heidelberg and the US sites, probably reflecting variation in folic acid fortification and supplement use.[13] Furthermore, ColoCare has published data on RECQ helicase expression,[37] NTRK3,[38] RET,[39] tumour-infiltrating lymphocytes and T cell receptor sequences,[40] 25–25-hydroxyvitamin $D_3$,[13 41] DNA methylation,[42–45] miRNAs,[46 47] faecal microbiota,[48–50] metabolomics and transcriptomics,[51–53] plasma proteins,[54] gene expression,[55] branched-chain amino acids,[56] genetic variants,[57] body composition,[53 58 59] physical activity[41] and dietary patterns[4] in patients with CRC. Within the COLON study, results have been published on body weight trajectories,[60] changes in lifestyle,[61] 25-hydroxy vitamin D levels[62 63] and inflammation markers[64] over time, as well as vitamin D,[64]

calcium or magnesium intake,[65] physical activity,[66] inflammation,[67 68] skeletal muscle mass[69]/density[70] and other measures of body composition[71] in relation to cancer recurrence, survival or physical functioning or fatigue. Moreover, in a subset of patients undergoing chemotherapy, dietary factors in relation to chronic CIPN[72 73] as well as chemosensory perception and food preferences[74] were studied. The Austrian CORSA Study has published results on genomic data,[75–77] telomere length,[78] DNA repair processes,[79] tumour autoantibodies[80] as well as metabolomics.[33 81] To date, publications from the EnCoRe Study have reported on associations of physical activity and sedentary behaviour,[82–85] adherence to lifestyle guidelines,[86] and parameters of body composition[87 88] measured through CT scans with quality of life, functioning and fatigue in CRC survivors. Recently, longitudinal associations between supplement use and fatigue were investigated from diagnosis to 2 years post-CRC treatment. No overall association between supplement use and fatigue was found but results suggest that increased levels of fatigue may be a reason for the use of supplements among CRC survivors.[89] Higher concentrations of 25OHD3 were longitudinally associated with better global quality of life and less fatigue in CRC survivors within EnCoRe suggesting a potential beneficial role of vitamin D in colorectal cancer survivors.[35] In a mixed-method study using data of the EnCoRe study, CRC (treatment) related health and functioning problems negatively impacted the ability of nearly one in five long-term CRC survivors to participate in everyday life situations and their satisfaction with participation.[90] The validity of the FFQ for measuring dietary intake among survivors of CRC within the EnCoRe study appeared to be moderate to good for most nutrients and food groups, relative to a 7-day dietary record.[91] Prediction models were developed for estimating 1-year risk of low health-related quality of life in seven domains in CRC survivors and performed well when externally validated among survivors within the EnCoRe and COLON studies.[92]

### Future plans
The consortium specified a comprehensive manuscript list of future projects using data from the FOCUS Consortium. Some selected projects are described below:
a. Recently, the investigation of longitudinal associations of adherence to the dietary World Cancer Research Fund/American Institute for Cancer Research and Dutch Healthy Diet recommendations with plasma kynurenine levels in CRC survivors after treatment has been finalised and the corresponding manuscript is under review at an international journal.
b. Further, near-term future plans include the investigation of (1) biomarkers related to FOCM and associations with folate intake (from diet and supplements); (2) associations between FOCM biomarkers such as vitamin $B_{12}$ and tryptophan and recurrence, survival, and patient-reported outcomes in CRC; (3) the impact of folate status (FOCM biomarkers and diet/

supplements) on treatment toxicity in patients treated with 5-floururacil modifiers; (4) the interaction between biomarkers related to FOCM and polymorphisms in FOCM-related genes in relation to CRC prognosis (recurrence and survival); (5) prognosis (disease-free and overall survival) in stages I–III CRC and associations with dietary and supplement use at diagnosis and changes during and after treatment; (6) FOCM-related biomarkers and their association with body composition in stages I–III patients with CRC; (7) associations between folate status (FOCM biomarkers and diet/supplement use) and recurrence, survival and patient-reported outcomes in young-onset CRC.

c. Long-term plans include the combination of additional biomarkers measured by the individual cohorts within the next years (eg, microbiome data, omics and multiomics data).

## Strengths and limitations

This is the largest consortium to date addressing the research question of folate and FOCM biomarkers in relation to survival, recurrence, treatment toxicity and health-related quality of life outcomes in patients with CRC. The cohorts included in the FOCUS Consortium are designed to enable future pooling of data.[22] For that reason, methodologies, time points and measurement instruments generally overlap, with each study presenting unique features such as additional blood collection at the 6-week follow-up time point within the EnCoRe Study and blood draws during chemotherapy within the COLON Study. The pooled sample size provides sufficient power to investigate subgroup analyses across patients with CRC, for example, within groups of patients who underwent 5-floururacil based chemotherapy or stratified by disease stage. Furthermore, the assessment of biomarkers related to FOCM are conducted at a single, state-of-the-art laboratory and the biological materials are processed and stored according to standardised operation protocols across all study sites, enabling precise and accurate measurements of FOCM biomarkers. While RCTs are the gold standard for establishing causality, the FOCUS cohort with its longitudinal design can contribute to establish causal relationships, with appropriate statistical analyses. Further, the FOCUS data include a time-varying exposure to dietary supplement intake for future studies to consider. The collection of the longitudinal data on dietary supplement intake, a key exposure, is essential to obtain meaningful estimates and thus required for developing recommendations and guidelines regarding dietary intakes among patients with CRC. The study population is predominantly based on European cohorts (90.4%) in countries that have not implemented mandatory folic acid fortification. This enables us to study a population where dietary intake and dietary supplement use determine differences in folate status, yielding information of direct relevance to cancer patients. However, the generalisability of results to populations that have introduced folic acid fortification, including the USA, might

therefore also be limited.[93] Performing sensitivity analyses by excluding countries without folic acid fortification (eg, Germany) or investigating analyses separately for Germany and the USA might help to address differences in fortification status. Moreover, patients were predominantly White, thus, it is not possible to address racial and ethnic minorities. Ethnicity/race is an important determinant of folate status and metabolism may be different between African Americans and Hispanics,[94] thus, recommendations should be limited to this current population. Future studies are warranted in diverse populations and compared with the FOCUS cohort.

Another limitation of the consortium includes differences in collection strategies of the baseline study time point across included studies: CORSA, COLON and EnCoRe are recruiting patients with CRC at diagnosis, preferably prior to any cancer treatment while ColoCare is recruiting patients prior to surgery (ie, neoadjuvant treatment might have occurred prior to baseline blood and data collection). Nevertheless, the ColoCare protocol requires that no blood draw occurs within 2 weeks of neoadjuvant or adjuvant chemotherapy, limiting the influence of ongoing treatments on blood biomarkers. In some cohorts, the timing with respect to adjuvant chemotherapy is less clearly defined and this will be carefully considered in sensitivity analyses. Furthermore, survivor bias—a type of selection bias—may be introduced as some patients died or did not complete follow-up questionnaires or did not provide blood samples. It is possible that patients who experience more severe toxicities, worse clinical outcomes, or worse health-related quality of life, are underrepresented among those patients who completed follow-up measurements.[4] Cohort studies such as the one presented here generate critical knowledge about preventable causes of disease. However, selection bias may affect estimates. This is particularly true for non-participation at follow-up that may depend on both the exposure and outcome. Within a review, Nohr *et al* showed a range of methods to quantify and adjust for selection bias. Even with limited data on nonparticipants and those lost to follow-up, it is possible to examine how effect estimates in a specific study may be biased by selection.[95] The likelihood for reverse causation is small in this prospective cohort, as the exposure measurements (blood folate levels and intake through diet/supplements) were collected before the outcome (survival, recurrence and quality of life) occurred. Therefore, these outcomes are unlikely to have influenced the exposure measurements. Given the robust follow-up in these cohorts for outcomes and data availability, future studies will be able to consider key confounders as well as predictors of recurrence and survival.

## Author affiliations
[1]Department of General, Visceral and Transplantation Surgery, University Hospital Heidelberg, Heidelberg, Germany
[2]Department of Epidemiology, Maastricht University, Maastricht, The Netherlands
[3]Department of Medicine, Vanderbilt University Medical Center, Nashville, Tennessee, USA

[4]Vanderbilt-Ingram Cancer Center, Nashville, Tennessee, USA
[5]Center for Cancer Research, Medical University of Vienna, Wien, Austria
[6]Division of Human Nutrition and Health, Wageningen University & Research, Wageningen, The Netherlands
[7]BEVITAL, Bergen, Norway
[8]Huntsman Cancer Institute, Salt Lake City, Utah, USA
[9]Department of Population Health Sciences, University of Utah, Salt Lake City, Utah, USA
[10]Department of Statistics and Operations Research, University of Vienna, Wien, Austria
[11]Fred Hutchinson Cancer Research Center, Seattle, Washington, USA
[12]Department of Medicine, University of Washington School of Medicine, Seattle, Washington, USA
[13]Genome Biology, European Molecular Biology Laboratory (EMBL), Heidelberg, Germany
[14]Surgical Department I, Städtische Kliniken Neuss, Lukaskrankenhaus GmbH, Neuss, Germany

**Acknowledgements** ColoCare Heidelberg: We thank all ColoCare study participants, the liquid biobank of the National Center for Tumor Diseases for storage of the blood samples according to the SOPs of the Biomaterialbank Heidelberg (BMBH), Michaela Magenreuter, Maria Thomalla-Starzl, Mareen, Dupovac, Marzena Knyssok-Sypniewski, Lin Zielske, Anett Brendel, Renate Skatula, and Marita Wenzel for laboratory work, Rifraz Farook, Dr. Werner Diehl and Inna Müller for data management and clinical follow-up data collection, Dr. Clare Abbenhardt-Martin, Susanne Jakob, Judith Kammer for patient recruitment and follow-up, and Dr. Petra Schrotz-King and Dr. Jürgen Boehm for study coordination. ColoCare Huntsman Cancer Institute: We thank all ColoCare study participants, the biorepository at Huntsman Cancer Institute for storage of the blood samples according to standardised SOPs, and Christy Warby for laboratory work. We thank Tyler Farr, Sarah Fischbuch and Debra Ma for data management and clinical follow-up data collection. We are very grateful for patient recruitment and follow-up by Elisa Santori, Hanna Omar-Payne and Marissa Grande, and Dr. Jürgen Boehm for study coordination. ColoCare Fred Hutchinson Cancer Research Center: We thank all ColoCare study participants, the biorepository at Fred Hutchinson Cancer Research Center for storage of the blood samples according to standardised SOPs and the laboratory team for sample processing. We thank the ColoCare team for patient recruitment and follow-up and data management. EnCoRe: We would like to thank all participants of the EnCoRe study and the health professionals in the three hospitals involved in the recruitment of participants of the study: Maastricht University Medical Center+, VieCuri Medical Center, and Zuyderland Medical Center. We would also like to thank the MEMIC center for data and information management for facilitating the logistic processes and data management of our study. Finally, we would like to thank the research dieticians and research assistant who are responsible for patient inclusion and follow-up, performing home visits, as well as data collection and processing. CORSA: We kindly thank all individuals who agreed to participate in the CORSA study. Furthermore, we thank Thomas Bachleitner-Hofmann, Michael M. Bergmann, Judith Karner-Hanusch and Anton Stift (Department of Surgery, Medical University of Vienna), Karl Mach, Azita Deutinger-Permoon (KRAGES, Austria), and their co-workers for supporting CORSA recruitment. In addition, we would like to acknowledge Elisabeth Feik, Philipp Hofer and Cornelia Zöchmeister, for patient recruitment, sample processing, and follow-up. COLON: The authors would like to thank the COLON participants, COLON investigators at Wageningen University & Research and the involved coworkers in the participating hospitals.

**Collaborators** A substantial amount of time has been spent in creating the harmonized dataset of baseline variables from cohorts within the FOCUS Consortium, with follow-up data collection and FOCM biomarker analyses. Any person interested in collaborating, learning about the FOCUS Consortium or in getting access to FOCUS data and in-depth analyses can contact the coordinating study PIs Prof. Andrea Gsur (andrea.gsur@meduniwien.ac.at) and Prof. Cornelia Ulrich (neli.ulrich@hci.utah.edu). Requests for data will be discussed and decided by all PIs and will require a Data Transfer Agreement.

**Contributors** BG, MW, EK, PMU, CU, AG, AU and NH: conceived of the project, developed the overall research plan, and provided study oversight; BG, EvR, ANH, SB, AJMRG, AU, JO, JLK, VD, RK, TG, TL, GK, TK, DEK, FJvD, MJB, AB, CIL, WG and KV: conducted hands-on experiments and data collection; BG, EvR, ANH, SB, AJMRG, AU, JO, JLK, VD, RK, TG, TL, GK, TK, DEK, FJvD, CIL, WG, KV and CU: provided essential reagents or essential materials, databases, and so forth, necessary for the research; BG, EvR, MJB, AB, ANH, VD and SB: analysed the data or performed the statistical analysis and made a major contribution to writing the paper; BG, MS, EK, PMU, AU, MW, AG and CU: had primary responsibility for the final content; CU is acting as guarantor; all authors: reviewed the manuscript critically, provided feedback and read and approved the final manuscript.

**Funding** The FOCUS Consortium is an ERA-NET (European Research Area Network) on Translational Cancer Research (TRANSCAN) project funded by the local research funding agencies (German Ministry of Education and Research, Germany, 01KT1503; Dutch Cancer Society with grant number UW2014-6877, the Netherlands; FWF Austrian Science Fund, Austria (API02104FW); Research Council Norway/Norwegian Cancer Society (RNC), Norway). The COLON study is further sponsored by Wereld Kanker Onderzoek Fonds, including funds from grant 2014/1179 as part of the World Cancer Research Fund International Regular Grant Programme; Alpe d'Huzes/Dutch Cancer Society (UM 2012-5653, UW 2013-5927, UW 2015-7946); and ERA-NET on Translational Cancer Research (TRANSCAN/ Dutch Cancer Society: UW2013-6397 and the Netherlands Organization for Health Research and Development (ZonMw), the Netherlands). The EnCoRe study was supported by grants from the Stichting Alpe d'HuZes within the research program 'Leven met kanker' of the Dutch Cancer Society (Grant No. UM-2010-4867 and UM-2012-5653), and by grants from Kankeronderzoekfonds Limburg as part of Health Foundation Limburg (Grant No. 00005739), and by a grant from Wereld Kanker Onderzoek Fonds (WKOF), as part of the World Cancer Research Fund International grant programme (grant number 2016/1620). BG was funded by the ERA-NET on Translational Cancer Research (TRANSCAN), the German Ministry of Education and Research project 01KT1503, the National Institutes of Health/ National Cancer Institute (NHI/NCI) projects R01 CA189184 and U01 CA206110, the Stiftung LebensBlicke, and the Matthias-Lackas Foundations. EvR was funded by the Wereld Kanker Onderzoek Fonds (WKOF), part of the World Cancer Research Fund International grant programme (grant number 2016/1620). Martijn Bours was funded by Alpe d'HuZes within the research program 'Leven met kanker' of the Dutch Cancer Society (Grant No. UM-2012-5653). DEK is supported by a Veni grant of the Netherlands Organisation for Scientific Research (grant no. 016. Veni.188.082). ANH was supported by the National Institutes of Health under Ruth L. Kirschstein National Research Service Award T32 HG008962 from the National Human Genome Research Institute. Nina Habermann was funded by the TRANSCAN project 01KT1512, Cornelia M. Ulrich and Jennifer Ose were funded by the Huntsman Cancer Foundation, and NIH/NCI projects R01 CA189184, R01 CA207371, U01 CA206110, U01 CA206110 and P30 CA042014. NIH grants (P30 CA15704, U01 CA152756, R01 CA194663, R01 CA220004, P01 CA077852), Rodger C. Haggitt Endowed Chair, R.A.C.E. Charities, Cottrell Family Fund, Listwin Family Foundation, Seattle Translational Tumor Research program, Fred Hutchinson Cancer Research Center (William M. Grady). Stefanie Brezina was funded by the ERA-NET on Translational Cancer Research (TRANSCAN) project FOCUS I2104-B26 and the ERA-NET on Translational Cancer Research (TRANSCAN) project MetaboCCC I 1578 – B19. JLK was supported by Kankeronderzoekfonds Limburg as part of Health Foundation Limburg (Grant No. 00005739). VD was funded by the German Ministry of Education and Research project (01KD2101D).

**Competing interests** None declared.

**Patient and public involvement** Patients and/or the public were not involved in the design, or conduct, or reporting, or dissemination plans of this research.

**Patient consent for publication** Not applicable.

**Ethics approval** The Heidelberg ColoCare Study was approved by the Ethics Committee of the Medical Faculty of Heidelberg (approval no. S-310/2001 and S-134/2016). The ColoCare Study at the Huntsman Cancer Institute and the Fred Hutchinson Cancer Research Center were approved by the respective IRBs (#77147 and #6407). For the CORSA study all subjects gave written informed consent and the study was approved by the institutional review boards: 'Ethikkommission Burgenland' (33/2010), by the ethical review committee of the Medical University of Vienna (1160/2016) and by the 'Ethikkommission der Stadt Wien' (EK 06-150VK). The EnCoRe Study was approved by the Medical Ethics Committee of the Maastricht University Hospital and Maastricht University (METC 11-3-075). Participants gave informed consent to participate in the study before taking part. Ethical approval for the study was granted by the Committee on Research involving Human Subjects, region Arnhem-Nijmegen under file number 2009/349.

**Provenance and peer review** Not commissioned; externally peer reviewed.

**Data availability statement** Data are available on reasonable request. Data described in the manuscript, code book and analytical code have been generated

from European-based consortia and as such are subject to regulations from multiple European countries, which limit our availability to share data. The consortium's funding has ended, and no centralised staff is available to support data requests. However, the FOCUS PIs have agreed to answer any queries or discuss potential projects with anyone interested in future collaborative research. For further questions, please contact colocarestudy_admin@hci.utah.edu.

**ORCID iDs**
Biljana Gigic http://orcid.org/0000-0002-8085-2072
Cornelia Ulrich http://orcid.org/0000-0001-7641-059X

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
