## [Reviewer comments · BMJ Open]

ARTICLE DETAILS

TITLE (PROVISIONAL)	Cohort profile: Biomarkers related to folate-dependent one-carbon metabolism in colorectal cancer recurrence and survival: The FOCUS Consortium
AUTHORS	Gigic, Biljana; van Roekel, Eline; Holowatyj, Andreana; Brezina, Stefanie; Geijssen, Anne; Ulvik, Arve; Ose, Jennifer; Koole, Janna L.; Damerell, Victoria; Kiblawi, Rama; Gumpenberger, Tanja; Lin, Tengda; Kvalheim, Gry; Koelsch, Torsten; Kok, Dieuwertje; van Duijnhoven, Franzel J; Bours, Martijn J.; Baierl, Andreas; Li, Christopher I.; Grady, William; Vickers, Kathy; Habermann, Nina; Schneider, Martin; Kampman, Ellen; Ueland, Per Magne; Ulrich, Alexis; Weijenberg, Matty; Gsur, Andrea; Ulrich, Cornelia

VERSION 1 – REVIEW

REVIEWER	Muliira, JK College of Nursing, Department of Adult Health and Critical Care Sultan Qaboos University
REVIEW RETURNED	31-Mar-2022

GENERAL COMMENTS	Dear Authors, Well done. The manuscript is well written and speaks of outcomes of folate which have been of concern in recently. I feel that the manuscript is comprehensive and provides a clear synthesis of the work done and opportunities to clarify the gaps through additional studies. However, in this manuscript I feel it is important explain the exact folate species that have been linked to inflammation and angiogenesis. There is also need to clarify or state clearly the differences in risk/bioavailability between folate intake from natural/nutrition sources versus supplements/adulterated.
--

REVIEWER	Alverdy, J University of Chicago
REVIEW RETURNED	06-May-2022

GENERAL COMMENTS	Please clearly state the aims of the current report, not the study you propose to do, but rather the actual aims of this current report: why are you presenting the design of a study rather than the study results itself? What is the purpose of the report? Please clarify how the results will likely end up as associations without causal inference of the role of folate, dietary supplements and CRC. The study, which only uses blood samples as they are easy to obtain and available as probabilistic NOT deterministic biomarkers. I found the scientific premise behind the proposed study to be weak, association not causation; like all serum based biomarker studies, probabilistic not deterministic.
--

REVIEWER	Étiévant , Lola NIH
REVIEW RETURNED	19-Jul-2022

GENERAL COMMENTS	The authors have submitted a “cohort profile” article, to present the FOCUS Consortium. They described its goals, participants, methods and findings to date. They also discussed some of its strengths and limitations. I have found the article well written and giving a good overview of the FOCUS Consortium. Because the non-statistical aspects of the manuscript were outside of my field of expertise, I have focused on the analyses mentioned by the authors in the Section “Key findings and publications”. The main purpose of the FOCUS Consortium is to study the associations between folate and folate-mediated one-carbon metabolism (FOCM) biomarkers and recurrence and survival in colorectal cancer (CRC) patients, to develop guidelines regarding folate intake among CRC patients. The authors have put the emphasis on findings presented in three published articles: Geijsen et al. (JNCI Cancer Spectrum, 2020), Kiblawi et al. (BJN, 2020) and Koole et al. (Am J of Clin Nutr, 2021). They stated that (i) an increased risk of cancer recurrences was observed among patients with higher compared to lower concentrations of circulating folic acid; (ii) specific folate species within the FOCM pathway were associated with inflammation and angiogenesis pathways among CRC patients; and (iii) higher vitamin B6 status was associated with better quality of life at six months post treatment. The detailed statistical analyses are given in the published articles, and have thus been previously reviewed. However, I do have some comments and questions. My main comment concerns the fact that to develop the guidelines, causal relationships between, e.g., folic acid and CRC recurrence must be established. However, the present analyses do not eliminate the threats of reverse causation and confounding bias. Even if certain limitations (e.g., selection bias) have been touched on by the authors, I do not think these two points have been clearly mentioned in the manuscript. The next points are mostly elaborations of this comment.  1. For example, for finding (i) (page 18/20 line 343), high concentrations of circulating folic acid may be due to an excessive intake of folic acid from dietary supplement and/or low activity of dihydrofolate reductase (Geijsen et al.). In addition, the use of supplements has been associated with a low quality of life and an increased fatigue (Koole et al.). However, I do not think that supplement intake, that could be an important confounder, has been adjusted for in the analysis. The observed association may therefore be biased, and in that case, its usefulness in establishing recommendations is limited. More generally, I think predictors of recurrence and survival should be adjusted for in such analyses. 2. In this same analysis (i), the exposure was assessed only through the blood sample drawn around the time of diagnosis, even though the relationship between the concentration of circulating folic acid and CRC recurrence may be more complex. In particular, the association between low quality of life, increased fatigue and the use of supplements could also indicate reverse causation.
---

3. In addition, if some CRC patients change their intake of dietary supplements, the exposure at baseline could differ from that at, say, six months. Why not use a “time-varying” exposure in analysis (i)? Is the intake of dietary supplements expected not to change after CRC diagnosis? Or was only the first sample used to limit the influence of cancer treatment?

4. From findings (i) and (ii), the authors suggest that B vitamins involved in the one-carbon metabolism may be associated with carcinogenic processes (p 18/20 | 349). However, to interpret causally such association, should not a unidirectional causal relationship (notably between folate species and inflammation and angiogenesis pathways) be assumed? Indeed, analysis (ii) was cross-sectional. Could the authors please comment on that?

I think the authors will need to answer the following questions to provide reliable guidelines: How could these limitations be overcome? In particular, how could the authors confirm the needed causal relationships? How will this be addressed in future plans? For example, could dietary/supplements interventions be performed on CRC patients? Overall, I understand that causal conclusions may be limited because of the nature of the data. But perhaps the authors could be upfront with the fact that biases due to reverse causation and unmeasured confounders can strongly impede the development of the recommendations, which is the main goal of the FOCUS Consortium.

Some minor comments:

1. For finding (ii) (p 18/20 | 348), are the directions of the associations relevant?

2. On p 21/23 | 421, the authors have raised concerns regarding the generalizability of their results, because the analyses have been conducted mostly in countries without mandatory acid folic fortification, unlike e.g., in the United States. How can this be addressed in the development of the guidelines? A similar concern could be raised if the CRC patients in the study were e.g., predominantly white. Was it the case? Could this have an impact on the findings and on the recommendations?

3. The authors have mentioned that certain protocols in the FOCUS Consortium try to limit the influence of cancer treatments on blood biomarkers (p 21/23 | 433). But if cancer treatment has an impact on folate, should not this be taken into account when establishing the recommendations for CRC patients?

4. The authors mentioned in their limitations (p 21/23 | 434) that cancer treatment might have occurred prior to collection of the first blood sample for some of the CRC patients. What are the possible implications? For example, even if chemotherapy status is adjusted for in analysis (i), could this strengthen or weaken the association that was found?

5. Similarly, how do the authors expect the possible selection bias to impact their results?

6. The authors have proposed an item “future plans” in the Abstract. This only provides a brief description, and I don’t think I

	have seen a dedicated Section in the manuscript. The authors did give a few leads here and there (e.g., p 21/23 434), but they could perhaps provide further details (e.g., planned statistical analyses).
--	--

VERSION 1 – AUTHOR RESPONSE

Reviewer #1

Dr. JK Muliira, College of Nursing, Department of Adult Health and Critical Care Sultan Qaboos University

Comments to the Author:

Dear Authors,
Well done.

The manuscript is well written and speaks of outcomes of folate which have been of concern in recently. I feel that the manuscript is comprehensive and provides a clear synthesis of the work done and opportunities to clarify the gaps through additional studies.

We thank the author for the positive feedback about our manuscript and for highlighting opportunities within this cohort to fill gaps via future studies.

1) However, in this manuscript I feel it is important explain the exact folate species that have been linked to inflammation and angiogenesis.

We agree with the reviewer and have added the following information to the manuscript:

Paragraph “Key findings and publications”, p. 19:

In particular, vitamin B6 species, pyridoxal 5'-phosphate (PLP), pyridoxal (PL) and pyridoxic acid (PA), were inversely associated with inflammatory biomarkers C-reactive protein (CRP), serum amyloid A (SAA), IL-6 and IL-8. Thiamine and thiamine monophosphate were inversely correlated with the CRP and IL-6. In addition, positive correlations of PA, PL and PLP with angiogenesis biomarker VEGF-D were observed.

2) There is also need to clarify or state clearly the differences in risk/bioavailability between folate intake from natural/nutrition sources versus supplements/adulterated.

We thank the reviewer for the comment and have revised the manuscript accordingly.

Paragraph “Introduction”, p. 3:

However, there is broad agreement that food folate is less bioavailable than folic acid with a median relative bioavailability of 65% (range: 44–80%), an estimate that approximates the 60% value derived from the Dietary Folate Equivalents equation.

Reviewer #2

Dr. J Alverdy, University of Chicago, University of Chicago Biological Sciences Division

Comments to the Author:

1) Please clearly state the aims of the current report, not the study you propose to do, but rather the actual aims of this current report: why are you presenting the design of a study rather than the study results itself? What is the purpose of the report?

We thank the reviewer for their feedback. The main purpose of this manuscript, a Cohort profile, is to inform the scientific community about our ongoing studies and the combined FOCUS Consortium data, to advise interested researchers of opportunities for collaboration, and to describe the complex methodology of a large consortium. This is typically done for large consortia or studies, outlines the

uniform methodology for all subsequent studies that utilize these data. An advantage to this article type is that all methodology for the cohort, including biomarker data, is available to the scientific community for secondary data analysis and research.

This cohort profile herein describes a large consortium of prospective studies with follow-up of patients and includes abstraction of key clinical outcomes such as recurrence and survival. The paper describes in detail the data collection methods as well as the main characteristics of the cohort. This goes beyond what is usually described in the methods section of a research paper. We revised the manuscript and added the purpose of the report.

Paragraph "Introduction", p. 4:

The main purpose of the FOCUS Cohort profile is to (1) inform the scientific community about the FOCUS Consortium, (2) describe the complex methodology of a large consortium, (3) present ongoing studies using this infrastructure as well as (4) advise interested researchers of opportunities for collaboration.

2) Please clarify how the results will likely end up as associations without causal inference of the role of folate, dietary supplements and CRC. The study, which only uses blood samples as they are easy to obtain and available as probabilistic NOT deterministic biomarkers. I found the scientific premise behind the proposed study to be weak, association not causation; like all serum based biomarker studies, probabilistic not deterministic.

We agree with the reviewer that the main focus of observational studies is to investigate associations and not to determine causality. In order to address causality, randomized controlled trials (RCTs) are the gold standard. However, various treatment strategies which we consider "evidence based" have never been subject to a prospective RCT (e.g., RCT assigning patients to either a smoking or a non-smoking group), as we would consider it unethical to assign patients to a group which we know is harmful.

Similarly, the FOCUS Consortium has the unique opportunity to provide observational evidence that is critically needed, before time-consuming and expensive clinical trials are undertaken. We recently published highly relevant findings on vitamin B6 status and related biomarkers as predictors of survival. This is exactly the type of work needed for the development of future RCTs. We revised the manuscript according to this comment and added following information to the manuscript.

Paragraph "Introduction", p. 4:

This joint research may lead to a better understanding of the role of folate- and FOCM-related mechanisms in the prognosis of CRC and be a precursor for data for future randomized controlled trials, which will be critical for the development of guidelines regarding folate intake among CRC patients.

Reviewer #3

Dr. Lola Étievant, NIH

Comments to the Author:

The authors have submitted a "cohort profile" article, to present the FOCUS Consortium. They described its goals, participants, methods and findings to date. They also discussed some of its strengths and limitations. I have found the article well written and giving a good overview of the FOCUS Consortium.

Because the non-statistical aspects of the manuscript were outside of my field of expertise, I have focused on the analyses mentioned by the authors in the Section "Key findings and publications". The main purpose of the FOCUS Consortium is to study the associations between folate and folate-mediated one-carbon metabolism (FOCM) biomarkers and recurrence and survival in colorectal cancer (CRC) patients, to develop guidelines regarding folate intake among CRC patients. The

authors have put the emphasis on findings presented in three published articles: Geijsen et al. (JNCI Cancer Spectrum, 2020), Kiblawi et al. (BJN, 2020) and Koole et al. (Am J of Clin Nutr, 2021). They stated that (i) an increased risk of cancer recurrences was observed among patients with higher compared to lower concentrations of circulating folic acid; (ii) specific folate species within the FOCM pathway were associated with inflammation and angiogenesis pathways among CRC patients; and (iii) higher vitamin B6 status was associated with better quality of life at six months post treatment. The detailed statistical analyses are given in the published articles, and have thus been previously reviewed. However, I do have some comments and questions.

We greatly appreciate these comments from the Reviewer about our manuscript, and for the recommendations made to further improve and clarify this article.

1) My main comment concerns the fact that to develop the guidelines, causal relationships between, e.g., folic acid and CRC recurrence must be established.

We thank the Reviewer for the comment. Please see the comment p. 4, Reviewer #2, 2).

2) However, the present analyses do not eliminate the threats of reverse causation and confounding bias. Even if certain limitations (e.g., selection bias) have been touched on by the authors, I do not think these two points have been clearly mentioned in the manuscript. The next points are mostly elaborations of this comment.

We thank the Reviewer for this comment. We revised the manuscript according to this comment and added the following information on reverse causation and confounding bias to the manuscript.

Paragraph “Strengths and limitations”, p. 23:

The likelihood for reverse causation is small in this prospective cohort, as the exposure measurements (blood folate levels and intake through diet/supplements) were collected before the outcome (survival, recurrence, and quality of life) occurred. Therefore, these outcomes are unlikely to have influenced the exposure measurements.

a) For example, for finding (i) (page 18/20 line 343), high concentrations of circulating folic acid may be due to an excessive intake of folic acid from dietary supplement and/or low activity of dihydrofolate reductase (Geijsen et al.). In addition, the use of supplements has been associated with a low quality of life and an increased fatigue (Koole et al.). However, I do not think that supplement intake, that could be an important confounder, has been adjusted for in the analysis. The observed association may therefore be biased, and in that case, its usefulness in establishing recommendations is limited. More generally, I think predictors of recurrence and survival should be adjusted for in such analyses.

We thank the reviewer for the comment. We agree, that dietary supplement use is a potential confounder with respect to analyses of FOCM biomarker concentrations. We would like to clarify that for the studies mentioned:

- Geijsen et al. performed subgroup analyses to assess potential effect measure modification by dietary supplement use, but did not further adjust the applied multivariate models.
- Further, Koole et al. adjusted the analyses for any supplement use (yes/no) in addition to other potential confounders.

We extended the manuscript according to this comment by additional information.

Paragraph “Strengths and limitations”, p. 23:

Given the robust follow-up in these cohorts for outcomes and data availability, future studies will be able to consider key confounders as well as predictors of recurrence and survival.

b) In this same analysis (i), the exposure was assessed only through the blood sample drawn around the time of diagnosis, even though the relationship between the concentration of circulating

folic acid and CRC recurrence may be more complex. In particular, the association between low quality of life, increased fatigue and the use of supplements could also indicate reverse causation. We thank the Reviewer for this comment. We agree with the Reviewer that due to the observational nature of the study it is difficult to differentiate between cause and effect. Our team has considered the challenge of potential confounding very carefully. We refer the Reviewer to these articles for the detailed discussion on potential alternate explanations, including reverse causation.

c) In addition, if some CRC patients change their intake of dietary supplements, the exposure at baseline could differ from that at, say, six months. Why not use a “time-varying” exposure in analysis (i)? Is the intake of dietary supplements expected not to change after CRC diagnosis? We thank the Reviewer for this suggestion. As the work by Geijsen and Koole et al. have already been published, we will consider this in future publications from our consortium. We have added information on “time-varying” exposures within this cohort to the manuscript.

Paragraph “Strengths and limitations”, p. 22:

Another advantage of this cohort is the ability to include a time-varying exposure on dietary supplement intake for future studies to consider.

d) Or was only the first sample used to limit the influence of cancer treatment? In some cohorts of the FOCUS Consortium the baseline sample was drawn after potential neo-adjuvant treatment, therefore analyses are adjusted for neo-adjuvant treatment.

e) From findings (i) and (ii), the authors suggest that B vitamins involved in the one-carbon metabolism may be associated with carcinogenic processes (p 18/20 | 349). However, to interpret causally such association, should not a unidirectional causal relationship (notably between folate species and inflammation and angiogenesis pathways) be assumed? Indeed, analysis (ii) was cross-sectional. Could the authors please comment on that?

We thank the Reviewer for this comment. Please see p. 4, Reviewer #2, 2).

3) I think the authors will need to answer the following questions to provide reliable guidelines: How could these limitations be overcome?

We thank the Reviewer for the questions raised. We have addressed them as seen below.

a) In particular, how could the authors confirm the needed causal relationships?

The chance for reverse causation in our prospective cohort is unlikely. All exposure measurements (blood folate levels and intake through diet/supplements) were conducted before the outcome (survival, recurrence, and quality of life) had occurred. Therefore, these outcomes cannot have influenced the exposure measurements.

b) How will this be addressed in future plans?

We than the reviewer for this comment. We thank the reviewer for this comment. Please see p. 4, Reviewer #2, 2).

c) For example, could dietary/supplements interventions be performed on CRC patients?

We agree with the Reviewer that dietary and/or supplement interventions could be performed on CRC patients. Observational studies are used primarily to identify risk factors and prognostic indicators for future RCT trials. Furthermore, observational studies are a feasible option where randomized, controlled trials would be impossible or unethical (withheld treatment, assign participants to a smoking group). However, more high-level evidence from observational studies is critical, particularly regarding the potential role of folate in increasing risk of colorectal cancer.

Please also see p. 4, Reviewer #2, 2). We extended the manuscript accordingly.

Paragraph "Introduction", p. 4:

This joint research may lead to a better understanding of the role of folate- and FOCM-related mechanisms in the prognosis of CRC and be a precursor for data for future randomized controlled trials, which will be critical for the development of guidelines regarding folate intake among CRC patients.

d) Overall, I understand that causal conclusions may be limited because of the nature of the data. But perhaps the authors could be upfront with the fact that biases due to reverse causation and unmeasured confounders can strongly impede the development of the recommendations, which is the main goal of the FOCUS Consortium.

We agree with the Reviewer. As outlined above, we think the chance for reverse causation is small in our prospective cohort, because all of the exposure measurements (blood folate levels and intake through diet/supplements) were conducted before the outcome (survival, recurrence, and quality of life) had taken place. Therefore, these outcomes cannot have influenced the exposure measurements. We revised the manuscript according to the reviewer's comment.

Paragraph "Strengths and limitations", p. 23:

Cohort studies such as the one presented here generate critical knowledge about preventable causes of disease. However, selection bias may affect estimates. This is particularly true for non-participation at follow-up that may depend on both the exposure and outcome. Within a review, Nohr et al. showed a range of methods to quantify and adjust for selection bias. Even with limited data on nonparticipants and those lost to follow up, it is possible to examine how effect estimates in a specific study may be biased by selection. The chance for reverse causation is small in this prospective cohort, as the exposure measurements (blood folate levels and intake through diet/supplements) were collected before the outcome (survival, recurrence, and quality of life) occurred. Therefore, these outcomes are unlikely to have influenced the exposure measurements.

4) Some minor comments:

a) For finding (ii) (p 18/20 | 348), are the directions of the associations relevant?

We thank the Reviewer for this question. We feel that the directions of the associations are relevant as this can help identify lifestyle factors that are modifiable to improve clinical and patient-reported outcomes (e.g. health-related quality of life, fatigue, etc.).

b) On p 21/23 | 421, the authors have raised concerns regarding the generalizability of their results, because the analyses have been conducted mostly in countries without mandatory acid folic fortification, unlike e.g., in the United States. How can this be addressed in the development of the guidelines?

We thank the reviewer for this valuable comment. One option could be to perform sensitivity analyses excluding countries without folic acid fortification (e.g. Germany) or to investigate analyses separately for Germany and the US. This and country specific guidelines on folic acid intake might be helpful to address differences in fortifications. We added an additional paragraph to the manuscript.

Paragraph "Strengths and limitations", p. 22:

Performing sensitivity analyses by excluding countries without folic acid fortification (e.g. Germany) or investigating analyses separately for Germany and the US might help to address differences in fortification status.

c) A similar concern could be raised if the CRC patients in the study were e.g., predominantly white. Was it the case? Could this have an impact on the findings and on the recommendations?

We thank the reviewer for this interesting comment. We revised the manuscript accordingly. Thank you for this very important comment. Patients were predominantly White and we added this as a limitation to the manuscript.

Paragraph “Strengths and limitations”, p. 22:

Moreover, patients were predominantly White, thus, it is not possible to address racial and ethnic minorities. Ethnicity/race is an important determinant of folate status, and metabolism may be different between African Americans and Hispanics, thus, recommendations should be limited to this current population. Future studies are warranted in diverse populations and compared with the FOCUS cohort.

5) The authors have mentioned that certain protocols in the FOCUS Consortium try to limit the influence of cancer treatments on blood biomarkers (p 21/23 | 433). But if cancer treatment has an impact on folate, should not this be taken into account when establishing the recommendations for CRC patients?

The Reviewer points to a very challenging topic, the question as to how folate supplementation and dietary factors can impact the efficacy and toxicity of antifolate drugs used in this population. This is precisely why the FOCUS consortium was developed, and upcoming publications from our group point to exactly this interaction. This is critical knowledge to be gained, as the use of drugs targeting folate one carbon metabolites is high, and extends beyond colorectal cancer.

6) The authors mentioned in their limitations (p 21/23 | 434) that cancer treatment might have occurred prior to collection of the first blood sample for some of the CRC patients. What are the possible implications? For example, even if chemotherapy status is adjusted for in analysis (i), could this strengthen or weaken the association that was found?

We thank the Reviewer for this comment. As this is a very large consortium that includes several cohorts of colorectal cancer patients, together with the proportion of neo-adjuvant treated patients at approximately 20%, we are able to perform stratified analyses or a subset analyses for those patients who were not treated. Heterogeneity in associations between biomarkers and clinical outcomes stratified or subset analyses for those patients who were not treated prior to blood sample collection.

Heterogeneity in associations between biomarkers and clinical outcomes stratified by neo-adjuvant chemotherapy status could be assessed using likelihood-ratio tests for the comparison of the model fit for logistic regression models. If differential associations are found, this requires careful interpretation. However, these findings are the most intriguing and important in the field of folate one-carbon metabolite biomarkers and cancer treatment/survivorship.

7) Similarly, how do the authors expect the possible selection bias to impact their results?

We thank the Reviewer for the comment. We revised the manuscript and added following information on selection bias.

Paragraph “Strengths and limitations”, p. 23:

Cohort studies such as the one presented here generate critical knowledge about preventable causes of disease. However, selection bias may affect estimates. This is particularly true for non-participation at follow-up that may depend on both the exposure and outcome. Within a review, Nohr et al. showed a range of methods to quantify and adjust for selection bias. Even with limited data on nonparticipants and those lost to follow up, it is possible to examine how effect estimates in a specific study may be biased by selection.

8) The authors have proposed an item “future plans” in the Abstract. This only provides a brief description, and I don’t think I have seen a dedicated Section in the manuscript. The authors did give

a few leads here and there (e.g., p 21/23 | 434), but they could perhaps provide further details (e.g., planned statistical analyses).

Thank you. We have extended the “Future plans” section within the abstract, see p. 2, Editorial Request #2.

We also elaborated upon this in the manuscript by including an additional section “Future plans” to the revised article, see p. 2, Editorial Request #3.

1

VERSION 2 – REVIEW

REVIEWER	Muliira, JK College of Nursing, Department of Adult Health and Critical Care Sultan Qaboos University
REVIEW RETURNED	14-Sep-2022

GENERAL COMMENTS	The authors have calibrated the manuscript for clarity. They have responded adequately to the queries.
--

REVIEWER	Étiévant , Lola NIH
REVIEW RETURNED	15-Sep-2022

GENERAL COMMENTS	I would like to thank the Authors for their point-by-point responses, additions to the manuscript and corrections. I also think that the slightly modified structure of the manuscript has improved its readability. Regarding the main points in my previous review: First, confounding bias was a concern in the earlier version of the manuscript. The Authors now state that confounders will be considered in the future analyses. Then, in my previous comments 2), 3) and 4), I mentioned the cross-sectional analyses that had been performed. I was thinking of reverse causation for the relationship between the exposure and other predictors of the outcome. Indeed, such variables are most likely time-varying, and the relationships between e.g., fatigue and circulating folic acid are probably not unidirectional over time. In addition, even in the absence of any time-varying confounders, when the time-varying nature of the exposures is overlooked, I think that the causal interpretation of the estimated quantities is very limited. While I agree with the Authors' response that RCTs are the gold standard, observational data may be used to answer causal questions too. However, it requires analyzing the data very carefully, and usually, it is helpful to imagine the RCTs one would ideally perform, and to clearly state the causal questions one would like to answer. Of course, sometimes a causal question simply cannot be answered with the present observational data. And I think that a point intervention would certainly not be considered in a RCT to study e.g., the effect of circulating folic acid on CRC recurrence.
--

	In the Section “Strengths and limitations”, the Authors now state that an advantage of this cohort is the longitudinal data on dietary supplement intake. Overall, I think this not an advantage but a requirement, as most of the causal questions of interest to the Authors cannot be answered with time-point measurements of the exposures. I would suggest putting slightly more emphasis on the fact that this is needed to obtain estimates that are meaningful and thus useful for developing the guidelines.
--	--

VERSION 2 – AUTHOR RESPONSE

Reviewer #1

Dr. JK Muliira, College of Nursing, Department of Adult Health and Critical Care Sultan Qaboos University

Comments to the Author:

1) The authors have calibrated the manuscript for clarity. They have responded adequately to the queries.

We thank Reviewer #1 for the positive feedback. Your comments were very helpful for the improvement of our article.

Reviewer #3

Dr. Lola Étiévant, NIH

Comments to the Author:

1) I would like to thank the Authors for their point-by-point responses, additions to the manuscript and corrections. I also think that the slightly modified structure of the manuscript has improved its readability.

We thank Reviewer #3 for the positive feedback and we very much appreciate your comments and recommendations which further improved this article.

Regarding the main points in my previous review:

2) First, confounding bias was a concern in the earlier version of the manuscript. The Authors now state that confounders will be considered in the future analyses.

Thank you for the positive feedback about the revised part on confounding bias.

3) Then, in my previous comments 2), 3) and 4), I mentioned the cross-sectional analyses that had been performed. I was thinking of reverse causation for the relationship between the exposure and other predictors of the outcome. Indeed, such variables are most likely time-varying, and the relationships between e.g., fatigue and circulating folic acid are probably not unidirectional over time. In addition, even in the absence of any time-varying confounders, when the time-varying nature of the exposures is overlooked, I think that the causal interpretation of the estimated quantities is very limited. While I agree with the Authors' response that RCTs are the gold standard, observational data may be used to answer causal questions too. However, it requires analyzing the data very carefully, and usually, it is helpful to imagine the RCTs one would ideally perform, and to clearly state the causal questions one would like to answer. Of course, sometimes a causal question simply cannot be answered with the present observational data. And I think that a point intervention would certainly not be considered in a RCT to study e.g., the effect of circulating folic acid on CRC recurrence.

Thank you for this comment. We agree with the Reviewer. We revised the manuscript based on this comment.

Paragraph "Strengths and limitations", p. 22:

While RCTs are the gold standard for establishing causality, the FOCUS cohort with its longitudinal design can contribute to establish causal relationships, with appropriate statistical analyses.

4) In the Section "Strengths and limitations", the Authors now state that an advantage of this cohort is the longitudinal data on dietary supplement intake. Overall, I think this not an advantage but a requirement, as most of the causal questions of interest to the Authors cannot be answered with time-point measurements of the exposures. I would suggest putting slightly more emphasis on the fact that this is needed to obtain estimates that are meaningful and thus useful for developing the guidelines.

We thank the Reviewer for this comment and we agree that the collection of longitudinal data on dietary supplement intake is a requirement rather than a strength. We revised the manuscript accordingly.

Paragraph "Strengths and limitations", p. 22:

Further, the FOCUS data includes a time-varying exposure on dietary supplement intake for future studies to consider. The collection of the longitudinal data on dietary supplement intake, a key-exposure, is essential to obtain meaningful estimates and thus required for developing recommendations and guidelines regarding dietary intakes among CRC patients.